# High Impedance Fault Detection in Medium Voltage Distribution Network Using Discrete Wavelet Transform and Adaptive Neuro-Fuzzy Inference System

**Veerapandiyan Veerasamy [1,\*], Noor Izzri Abdul Wahab [1] , Rajeswari Ramachandran [2], Muhammad Mansoor [1,3], Mariammal Thirumeni [4] and Mohammad Lutfi Othman [1]**

1. Advanced Lightning and Power Energy System (ALPER), Department of Electrical and Electronics Engineering, Faculty of Engineering, Universiti Putra Malaysia (UPM), UPM Serdang 43400, Selangor, Malaysia; izzri@upm.edu.my (N.I.A.W.); mansoor.upm@gmail.com (M.M); lutfi@upm.edu.my (M.L.O.)
2. Department of Electrical Engineering, Government College of Technology, Coimbatore 641013, Tamilnadu, India; rreee@gct.ac.in
3. Pakistan Institute of Engineering and Technology, Multan 59060, Pakistan
4. Department of Electrical Engineering, Rajalakshmi Engineering College, Chennai 602105, Tamilnadu, India; mariammal.t@rajalakshmi.edu.in
* Correspondence: veerapandian220@gmail.com; Tel.: +60-1133375102

**Abstract:** This paper presents a method to detect and classify the high impedance fault that occur in the medium voltage (MV) distribution network using discrete wavelet transform (DWT) and adaptive neuro-fuzzy inference system (ANFIS). The network is designed using MATLAB software R2014b and various faults such as high impedance, symmetrical and unsymmetrical fault have been applied to study the effectiveness of the proposed ANFIS classifier method. This is achieved by training the ANFIS classifier using the features (standard deviation values) extracted from the three-phase fault current signal by DWT technique for various cases of fault with different values of fault resistance in the system. The success and discrimination rate obtained for identifying and classifying the high impedance fault from the proffered method is 100% whereas the values are 66.7% and 85% respectively for conventional fuzzy based approach. The results indicate that the proposed method is more efficient to identify and discriminate the high impedance fault from other faults in the power system.

**Keywords:** Discrete Wavelet Transform (DWT); adaptive neuro-fuzzy inference system (ANFIS); fuzzy logic system (FLS); high impedance fault (HIF)

## 1. Introduction

The design of protective devices has been a great challenge for power system engineers to ensure the reliability and security of a power system. To achieve this, the protection equipment or components in power system need to be designed for accurate detection and classification of fault in the system. The various abnormalities that occur in electrical distribution networks are capacitor switching, high impedance faults, line faults and sudden load rejection and so on. Among these disturbances, the detection of high impedance faults on electrical power system networks have been one of the most challenging phenomenon faced by the today's electric utility industry [1]. Over the years, the typical protection schemes used to detect the fault in the system involves only the low impedance faults (i.e., the fault with infinitesimal low resistance) and it outperforms to locate the faults. On the flipside, the resistance of the fault path is very high when one of phases of the transmission line makes electrical contact with a semi-insulated object such as tree, pole, surface of the road, gravels,

concrete, dry land etc., which is called high impedance fault (HIF). The significance of HIF is the magnitude of fault current, ranging from 0 to 75 amperes and exhibits the arcing and flashing at the point of contact, leading to serious threat of electrical shock or fire to the public. Hence, the detection is more important from the public and reliable operation point of view [1,2]. To mitigate such crisis, some of the conventional schemes such as minimum reactance approach, voltage and current pattern of the system and the transients associated with these waves were used to locate the faults in the system [3–6].

Researchers have presented a large number of high impedance fault identification algorithms using a combination of computational intelligence methods such as Artificial Neural Network (ANN), Fuzzy, harmonic component analysis using Extreme Learning Machine (ELM), decision tree approach, Bayes classification, nearest neighbor rule approach etc., along with the signal processing techniques such as Haar transform, Stockwell transform, Fourier and Wavelet transform analysis for identification of fault in the system [7–17]. On the other side, the advancement in information and communication technologies were used by the researchers to transfer the fault data or information using wireless sensor, communication links and phasor measurement units to locate the fault in the system thereby reducing the blackout of the system but these approaches failed to identify HIF in the system [18–20]. Among all the methods presented in the literature, the wavelet analysis outperforms for identifying the fault in the system due to its variable window sizing, strong local analysis of frequency band, prone to aliasing effect and energy leaking [21]. For these reasons, the wavelet analysis is used in this paper for feature extraction to classify the faults using ANFIS system. The ANFIS was proposed by Jang based on Takagi-Seguno, the performance was superior than the Fuzzy and ANN methods, as it integrates the merits of both these methods by incorporating the neural network approach to fuzzy logic at each step thereby giving high prediction accuracy. Due to these merits, this approach has been extensively used as a classifier in various applications such as image processing, biomedicine application and fault in gear system with high accuracy and practicability [22].

The core aim of this research, presents a design of HIF model, its location and classification in MV radial distribution network using the combination of DWT with ANFIS. To accomplish this, the three phase fault current signals have been measured to obtain the feature extraction namely standard deviation (SD) value. The obtained SD value is used to train the ANFIS system for classifying the different types of faults in the system, contrary to use energy values for training. The energy value based training approach consumes more computer memory and due to which a delay may be introduced.

The paper is organized as follows: Section 2 presents the system model studied and background of wavelet analysis. Section 3 explains the proffered method of fault classification and discrete wavelet transform. Section 4 describes the Intelligence method such as Fuzzy and ANFIS method as a classifier. The results and discussion were presented in Section 5 and finally, the conclusion is made in the last part of the paper.

## 2. System Modelling

The distribution power system network, which is given in Figure 1, was used for analyzing the various faults that occur in the system. The system was developed in MATLAB/Simulink software environment with the introduction of various faults such as symmetrical, unsymmetrical and high impedance fault. It consists of: Grid source (50 MVA/30 KV), Distribution Transformer (12 MVA, 30 KV/13.8 KV), a common bus of MV (13.8 KV) distribution network and five number of radial type distribution feeders with integration of load facility [2]. Moreover, a simplified Emanuel two-diode model is considered for the analysis of HIFs as shown in Figure 2 and which consists of two variable DC voltage sources of 1 to 10 kV connected to diodes through a non-linear resistor of about 50 to 500 ohms, which lead to the non-linear arc characteristics [8].

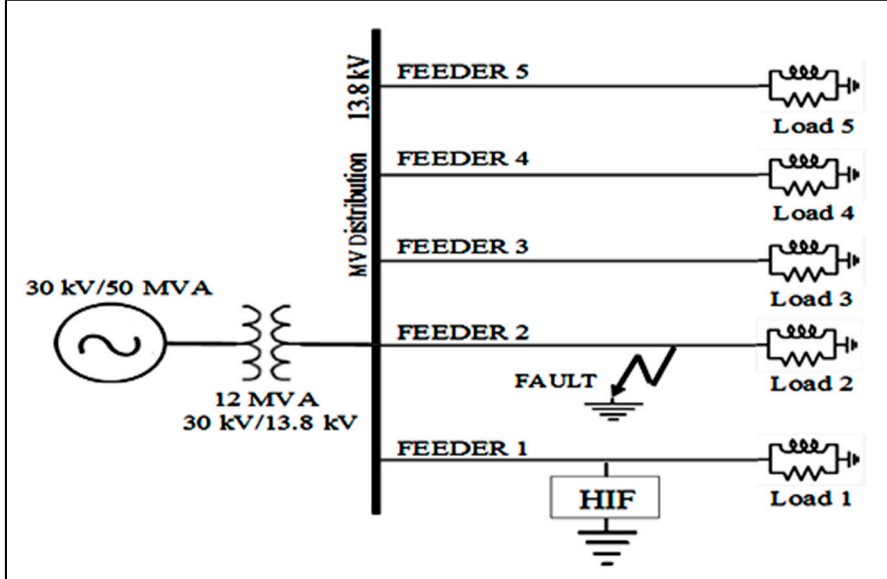

**Figure 1.** Radial distribution network of 13.8 kV.

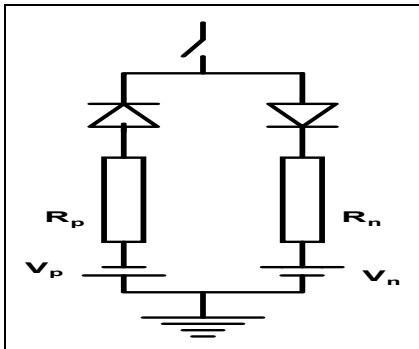

**Figure 2.** High impedance fault model.

*Background of Wavelet Analysis*

The transform theory such as Fast Fourier Transform (FFT) and Short Time Fourier Transform (STFT) technique gives the information about the signal localized only in frequency, the time at which particular disturbance or fault occurred in the signal is lost. Therefore, to overcome such drawbacks wavelet analysis were used in the proposed work to localize the signal both in time and frequency simultaneously which helps to detect the time of occurrence of fault or disturbances effectively.

Wavelet Transform is an effective mathematical tool used to analyze the signal with transients or discontinuities such as the post-fault voltage or current waveform. The wavelet transform uses the basis function of wide functional form and has features such as short windows at high frequencies and long windows at low frequencies as shown in Figure 3. A number of different wavelets have been used to approximate any given function with each wavelet being generated from one original wavelet, called a mother wavelet. The new elements formed called daughter wavelets, are scaled (dilated) and translated (time shifted) versions of the original wavelet.

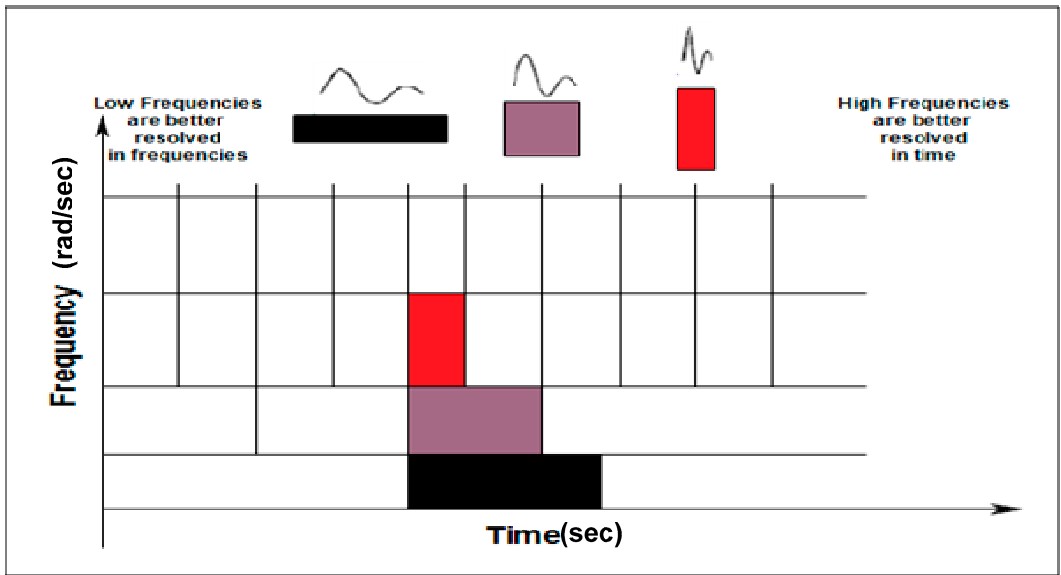

**Figure 3.** Wavelet Analysis.

The dilated or compressed form of mother wavelet implies scaling and shifting of mother wavelet in the time domain is called translation. The wavelet transform can be analyzed in two ways such as Continuous Wavelet Transform (CWT) and Discrete Wavelet Transform (DWT). Among these two methods, the latter is extensively used because of the following reasons [23,24],

- CWT requires a large number of scales to show the signal components, which makes it useless for online application.
- CWT is highly redundant transform as its wavelet coefficients contain more information than necessary.
- CWT provides the region where the fault occurs, but DWT localize the fault more efficient.
- DWT preserve all the information of the function with minimum number of wavelet coefficients.
- Computational time is faster for DWT analysis.
- Construction of CWT inverse is more difficult.

Due to the above disadvantages of CWT, in this proposed work the DWT based signal processing technique was used for location of fault in the power system.

## 3. Proposed HIF Detection Methodology

This section presents the methodology to detect the HIF in a MV distribution network as well as distinguish them from other faults in the system which comprises of 2 stages.

Initially (pre-processing Stage), the three phase current signal obtained using MATLAB simulation of MV radial distribution system is analyzed using wavelet analysis to obtain the feature extraction or data for training. In the second (Classification) stage, the ANFIS is trained to classify the state of the feeder network. The structure of proposed scheme using the application of wavelet transform and ANFIS to identify the faults is represented in Figure 4 by a simple block diagram describing the various stages and which is explained in detailed as below,

*Step 1—Pre-processing:* The fault current is obtained by simulating the MV distribution network with various faults in the power system.

*Step 2—Processing:* The original fault current signal is extracted from noise by decomposing the signal using DWT at various levels.

*Step 3—Feature Extraction:* The standard deviation (SD) for location of fault is extracted using 5-level DWT.

*Step 4*—**Traning:** The extracted SD values for various cases were used for training ANFIS system.

*Step 5*—**Classification:** Trained ANFIS based classifier algorithm identify the type of fault that occurs in the system.

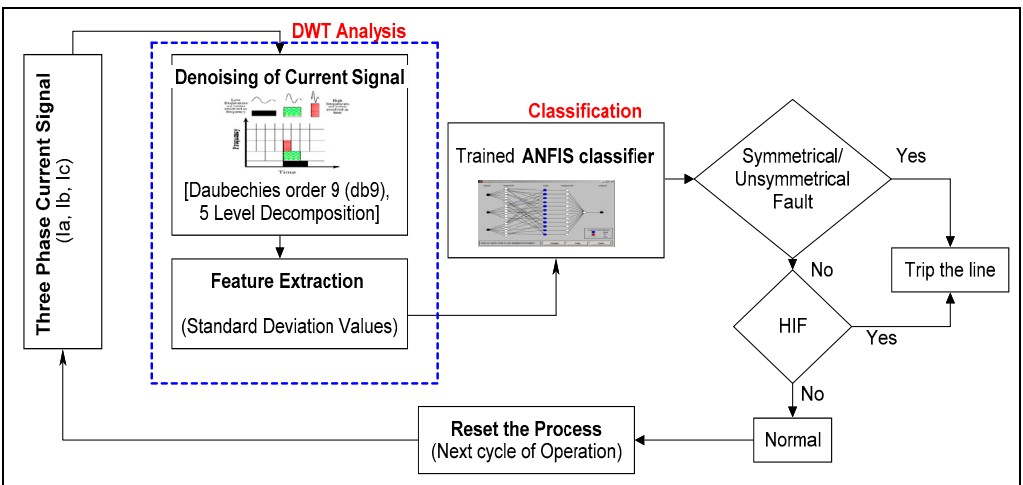

**Figure 4.** Schematic representation of proposed method.

### 3.1. Discrete Wavelet Transform

The DWT is a powerful time—frequency signal processing information tool which allows the signal to be sampled with localized transients and produces non-redundant restoration of signal. Moreover, it produces better spatial and spectral localization of signal. In recent decades, such advanced powerful tool has been used for designing the protective relays. In DWT, the fault current signal $x(t)$ is decomposed into low and high frequency components such as approximation (A) and detailed coefficients (D) as shown in underlying Equations (1)–(4) [25–29],

$$x(t) = \sum_k cA_0 \Phi_{j,k}(t) = \sum_k cA_1 \Phi_{j-1,k}(t) + \sum_k cD_1 \Phi_{j-1,k}(t) \quad x(t) = A_1(t) + D_1(t) \tag{1}$$

The low frequency component of the signal i.e., approximation coefficients undergoes the decomposition up to $N$ level called decomposition level to extract the original information from the noise and it is given in equation as below,

$$x(t) = A_2(t) + D_2(t) + D_1(t) \tag{2}$$

$$x(t) = A_3(t) + D_3(t) + D_2(t) + D_1(t) \tag{3}$$

In general the signal is represented as,

$$x(t) = A_N(t) + D_N(t) + D_{N-1}(t) + \ldots + D_1(t) \tag{4}$$

where $N$ is the decomposition level and the optimal decomposition with $L$ levels is allowed under the condition as given below (5),

$$N = 2^L \tag{5}$$

During the process of signal decomposition, the fault current signal at each level has been divided into different frequency bands as defined below,

$$B = \frac{F}{2^{L+1}} \tag{6}$$

where '$B$' is the bandwidth of each level in Hz and '$F$' is the Sampling Frequency in Hz.

### 3.1.1. Choice of Mother Wavelet

Selection of mother wavelet has been done by determining the decomposition at best level using Equations (5) and (6). The mother wavelet plays a major role and it depends on sampling frequency of the signal and frequency band of each levels. The sampling frequency acts as a filter at each level to find the best level of decompositions. Moreover, mother wavelet also depends on the type of application. In this study, detecting and analyzing low amplitude, short duration, fast decaying oscillating type of high frequency fault current signal DWT has been applied. One of the most popular mother wavelet of Daubichies's wavelet (Db9) has been used for wide range of application [30]. In this proposed work of fault analysis, sampling frequency of 20 kHz is considered for decomposing the signal into different levels with 5000 points in length for each phase of current signal. The band of frequencies captured for each level is varied and it is calculated using Equation (6) as shown in Table 1. The transients present at each subsequent level decreases compared to the previous level. The results have shown the transients were completely eradicated at level 5.

The mother wavelet of db9 has been used, because of its high Peak Signal to Noise Ratio (PSNR) compared to other mother wavelet of db4, db5, db7, and db8. In addition, it has the advantage of efficient way of reconstructing the original signal from sampled signal without loss of information.

**Table 1.** Frequency band of different detail coefficients.

| Detailed Coefficient Levels | Frequency Band kHz |
| --- | --- |
| D1 | 5 to 2.5 |
| D2 | 2.5 to 1.25 |
| D3 | 1.25 to 0.625 |
| D4 | 0.625 to 0.3125 |
| D5 | 0.3125 to 0.15625 |

### 3.1.2. Feature Extraction

This section presents the feature extraction to identify the type of fault that occurs in the system. The feature or training data called standard deviation (SD) of fault current signal ($x_i$) is obtained for each phase under different fault conditions using Equation (7) as given below,

$$\text{SD} = \sqrt{\frac{1}{n-1}\left[\sum_{i=1}^{n}(x_i - \bar{x})^2\right]} \tag{7}$$

where, $\bar{x} = \frac{1}{n}\sum_{i=1}^{n} x_i$, $x$ is the data vector and $n$ is the number of elements in that data vector.

The extracted SD values for each phase under various fault conditions with different value of fault resistance from the detailed and approximation coefficients of DWT is used for training the intelligence based classifier such as fuzzy logic system (FLS) and ANFIS for classification of different types of fault and to locate the HIF.

## 4. Intelligence-Based Classifier

This section presents the intelligence method of classifying the faults that occurs in power system. The intelligence techniques used in this paper are Fuzzy logic and ANFIS method to identify the HIF fault as described below,

### 4.1. Fuzzy Logic System

Fuzzy Logic System (FLS) is a form of many-valued logic with a set of common-sense rules. Furthermore, it is flexible, tolerant to imprecise data and natural language. Hence, this method is used to classify the type of fault that occurs in the system. The steps to be followed for training FLS is given in Figure 5 and also detailed as below [31,32],

**Step 1:** Define the problem and classify the data i.e., SD values

**Step 2:** Define the input and output fuzzy sets with variable name.

**Step 3:** Define the type of member function for each variable.

**Step 4:** Frame the rules.

**Step 5:** Built and test the system.

**Step 6:** Tune and validate the system.

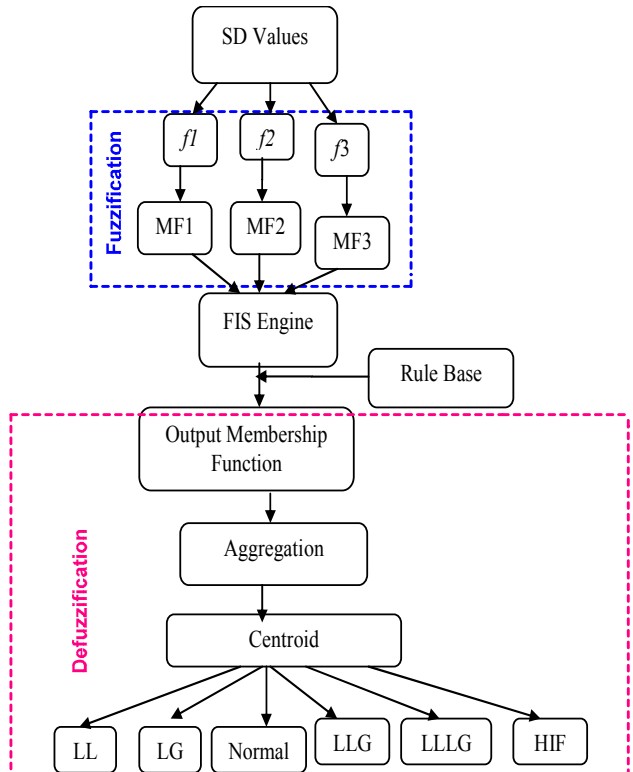

**Figure 5.** Fuzzy logic system (FLS) structure.

The fuzzy system has 3 inputs (S1, S2, and S3) and 1 output. For each phase, 4 triangular membership functions have been chosen such as HIF, ground, normal and fault. The range of values selected for each membership functions are 8 to 16 for HIF, 17 to 24 for normal, 25 to 35 for ground and 27 to 47 for fault. The output membership function has 6 triangular memberships function named as Normal, HIF, three phase fault, LL, LG and LLG fault.

The fuzzy rules are framed using S1,S2and S3 as phase A, Phase B and phase C are shown below,

- If (S1 is normal) and (S2 is normal) and (S3 is normal) then (trip output is Normal)
- If (S1 is fault) and (S2 is fault) and (S3 is fault) then (trip output is ABC fault)
- If (S1 is ground) and (S2 is ground) and (S3 is normal) then (trip output is ABG fault)
- If (S1 is normal) and (S2 is ground) and (S3 is ground) then (trip output is BCG fault)
- If (S1 is ground) and (S2 is normal) and (S3 is ground) then (trip output is ACG fault)
- If (S1 is ground) and (S2 is normal) and (S3 is normal) then (trip output is AG fault)
- If (S1 is normal) and (S2 is ground) and (S3 is normal) then (trip output is BG fault)
- If (S1 is normal) and (S2 is normal) and (S3 is ground) then (trip output is CG fault)
- If (S1 is fault) and (S2 is fault) and (S3 is normal) then (trip output is AB fault)
- If (S1 is normal) and (S2 is fault) and (S3 is fault) then (trip output is BC fault)
- If (S1 is fault) and (S2 is normal) and (S3 is fault) then (trip output is AC fault)
- If (S1 is HIF) and (S2 is normal) and (S3 is normal) then (trip output is HIF fault at Phase A)

- If (S1 is normal) and (S2 is HIF) and (S3 is normal) then (trip output is HIF fault at PhaseB)
- If (S1 is normal) and (S2 is normal) and (S3 is HIF) then (trip output is HIF fault at Phase C)

### 4.2. Adaptive Neuro Fuzzy Inference System

Adaptive Neuro Fuzzy Interface System (ANFIS) is one of the greatest tradeoff among ANNs and fuzzy logic systems, offering smoothness due to the fuzzy control interpolation and adaptability due to the ANN back propagation. ANFIS provide a technique for the implementation of fuzzy inference system to adaptive networks for developing fuzzy rules with proper membership functions to have required inputs and outputs. An adaptive network is a feed-forward multi-layer neural network with adaptive nodes in which the outputs are predicted on the parameters of the adaptive nodes and the adjustment of parameters due to error term is specified by the learning rules. ANFIS is a class of ANN, which incorporates both ANN and fuzzy logic principles and has benefits of both techniques in a single framework as follows [32–36]:

- It is capable of handling complex and nonlinear problems even if the targets are not given.
- The learning duration of ANFIS is very short than Neural Network (NN) which implies that ANFIS reaches the target faster than neural network.
- Reduces the complexity of the problem, in case of system with huge amount of data.
- In training of the data, ANFIS gives result with minimum total error compared to other type of NN.

ANFIS is an intelligent adaptive data learning method which maps the input and output through the input and output member function. From the input-output data, ANFIS adjusts the membership function using least square method or back propagation descent method for linear and non-linear system. The Sugeno fuzzy model has been proposed for creating the fuzzy rules from a given input-output data set. A typical Sugeno fuzzy rule is expressed in the following form [3,4]:

IF $\quad x_1$ is $A_1$
AND $\quad x_2$ is $A_2$
AND $\quad x_m$ is $A_m$
THEN $y = f(x_1, x_2, \ldots, x_m)$

where $x_1, x_2, \ldots, x_m$ are input variables; $A_1, A_2, \ldots, A_m$ are fuzzy sets. When $y$ is a constant, a zero-order sugeno fuzzy model is obtained in which the subsequent of a rule is specified by a singleton. When $y$ is a first-order polynomial equation, (i.e.,) $y = k_0 + k_1{}^*x_1 + k_2{}^*x_2 + \ldots + k_m{}^*x_m$, a first-order sugeno fuzzy model is obtained. The following Figure 6 illustrates the ANFIS structure with 6 layers.

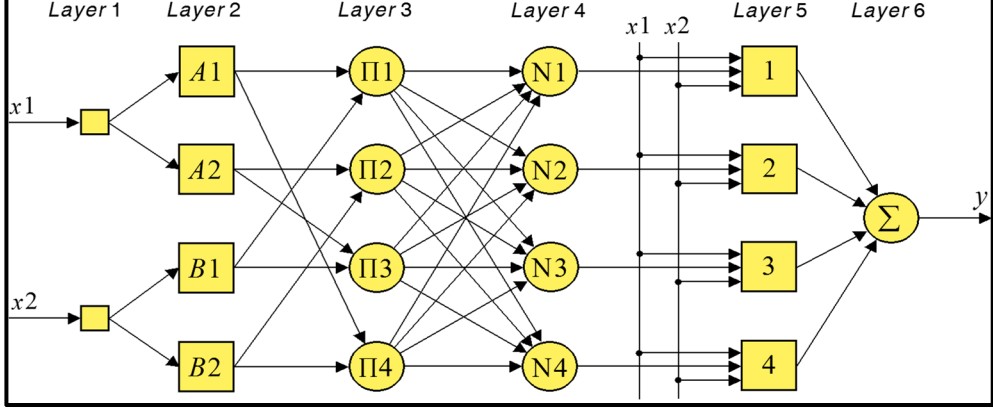

**Figure 6.** Adaptive Neuro Fuzzy Interface System (ANFIS) structure.

The 6 Layers in the ANFIS structures are input layer, fuzzification layer, rule layer, normalisation layer, defuzzification layer and output layer respectively. The input to the ANFIS detection model are standard deviation values of each phase of three phase system obtained by DWT analysis of fault current signal simulated by MATLAB model of radial distribution network. The output of ANFIS is the type of fault that occurs in the system.

The network is trained for the input-output data set with MATLAB ANFIS editor, which adjusts the MFs directly based on the data set. Four variables with triangular membership function are assigned for each input variable and the output is chosen to be constant because of its sugeno model. Fourteen rules are framed using FIS and 45 input–output data set is used for training ANFIS. The trained output data are shown in Table 2.

**Table 2.** Trained output data for ANFIS.

| S.No | Fault Type | Assigned Output |
|------|-----------|-----------------|
| 1 | No fault | 0 |
| 2 | HIF in phase C | 0.2 |
| 3 | HIF in phase B | 0.3 |
| 4 | HIF in phase A | 0.4 |
| 5 | LLL-G | 0.5 |
| 6 | LG (AG) | 0.6 |
| 7 | LG (BG) | 0.7 |
| 8 | LG (CG) | 0.8 |
| 9 | LL (AB) | 0.9 |
| 10 | LL (BC) | 1.0 |
| 11 | LL (AC) | 1.1 |
| 12 | LLG (ABG) | 1.2 |
| 13 | LLG (BCG) | 1.3 |
| 14 | LLG (ACG) | 1.4 |

The obtained output result of 0.2 in MATLAB/Simulink ANFIS structure for trained data indicates the type of fault that occurs in the distribution system is HIF fault as shown in the Figure 7.

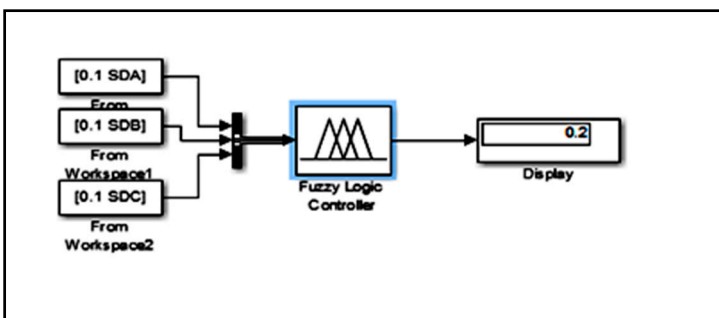

**Figure 7.** ANFIS results for distribution network.

## 5. Results and Discussion

### 5.1. Matlab Simulation Results for Different Cases

The radial distribution network system model of 13.8 kV with five distribution feeders for various fault study is located in Basque country (Spain) which is shown in Figure 1 is developed using MATLAB. The HIF fault model designed using MATLAB as depicted in Figure 8 consists of saw tooth current controller, constant resistor, variable resistor of non-linear fashion and diodes. The developed model has better dynamic arc current characteristics which depicts the non-linearity of ground resistance. The presented HIF model, symmetrical and asymmetrical fault is applied to the feeder network and

the current waveform is captured for fault time period of 0.02 to 0.08 s to study the type of fault that occurs in the system.

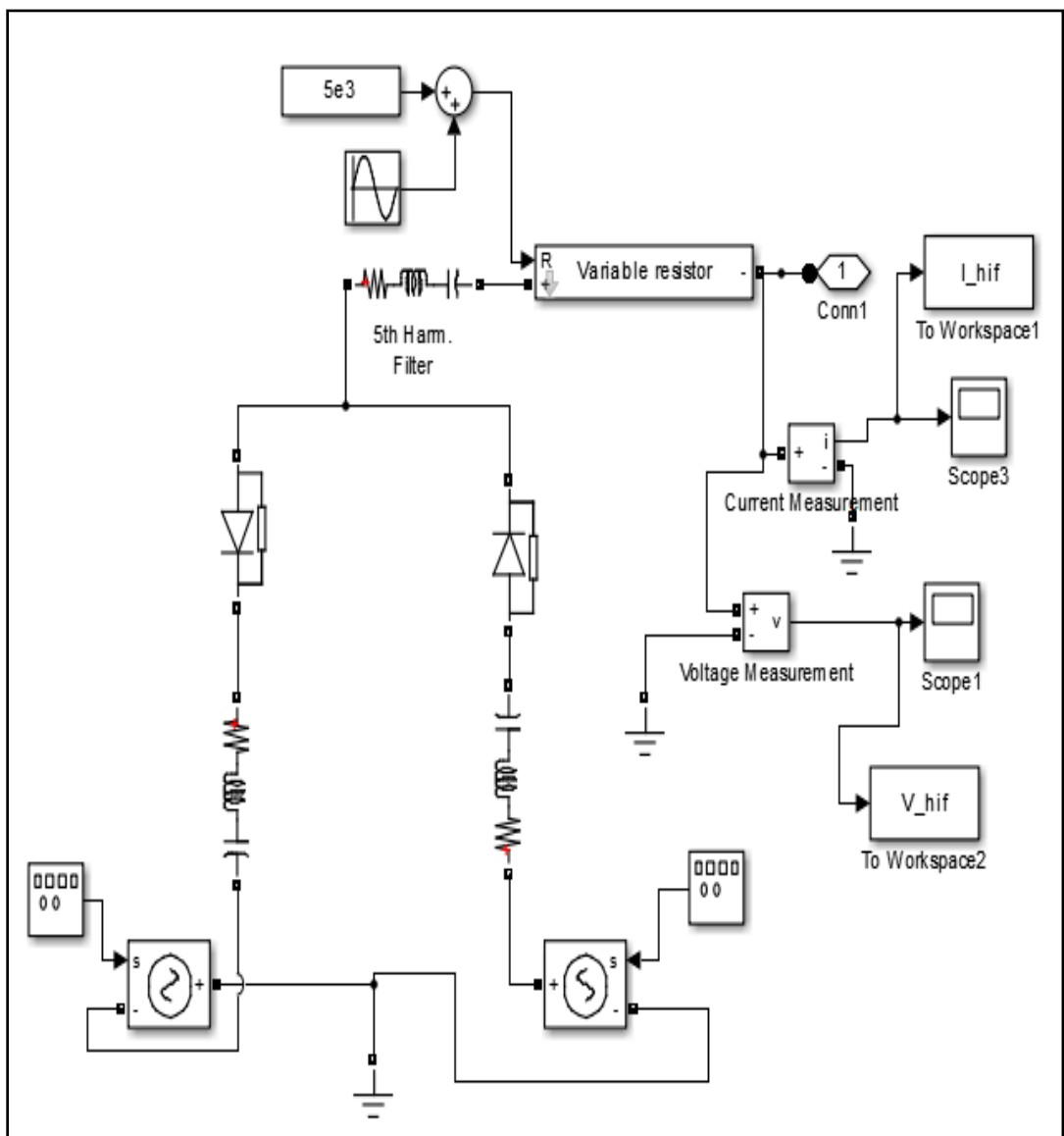

**Figure 8.** High impedance fault Matlab model.

　　The three phase current waveform measured in p.u under normal condition is shown in Figure 9 used as a reference signal to identify the disturbances or abnormalities that occurs in the system. The several faults such as Line to Ground (LG), Line to Line (LL),Double Line to Ground (LLG), three phase (LLLG), and high impedance fault (HIF) are applied to the feeder network and corresponding three phase current in p.u during fault is shown in Figures 10–14 respectively. It is observed that the transient or high frequency noise occurs during clearance and occurrence of fault. Moreover, it is seen that the magnitude of fault currents (p.u) in the case of LG, LL, LLG, and LLLG is very high but the magnitude of fault current (p.u) in case of HIF is low and it seems to be slightly higher than the normal three phase current of the system and the magnitude of HIF current in the Phase C of one of the feeder network is shown in Figure 15. Therefore, it is more challenging to detect this type of fault by conventional methods of fault protection scheme. This challenge has been addressed by the classifier method developed in this research.

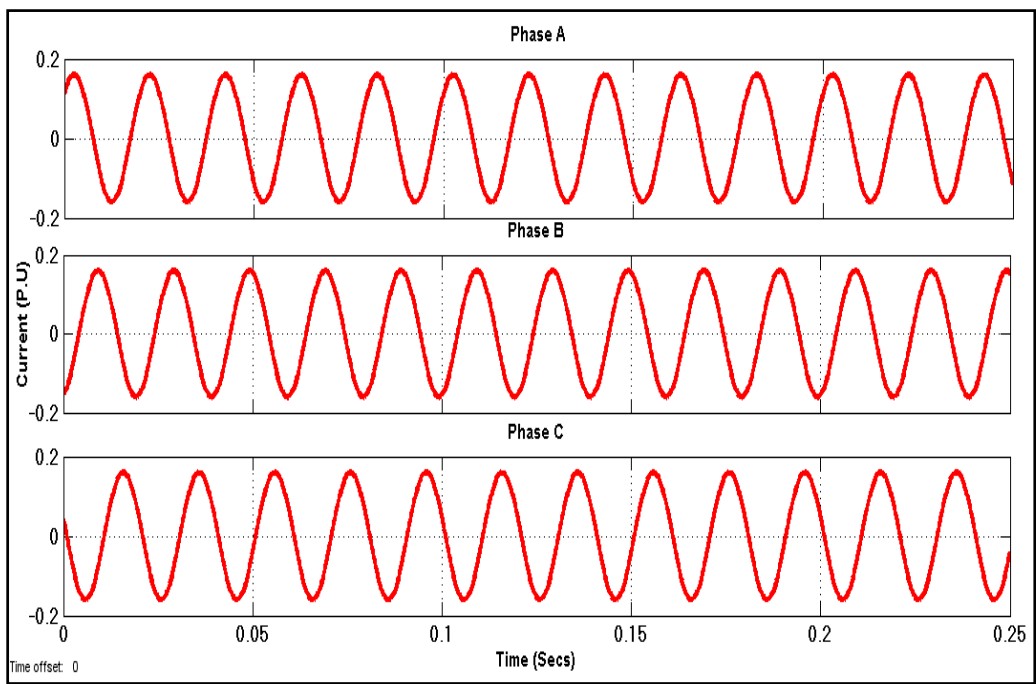

**Figure 9.** Three phase current waveform for normal case.

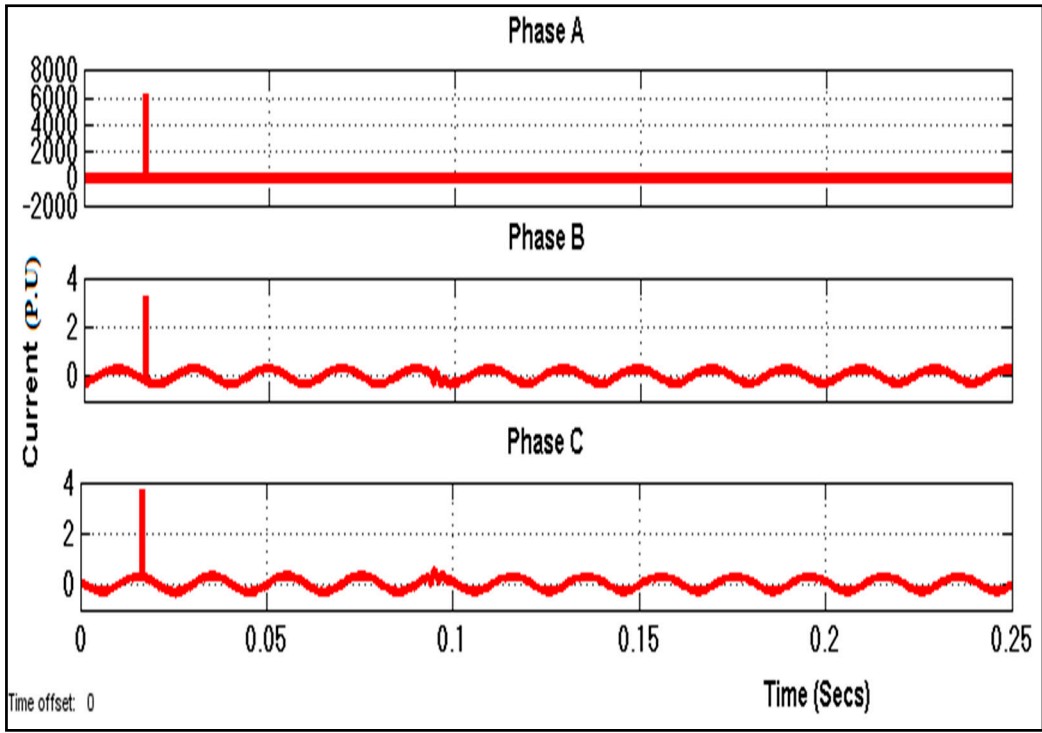

**Figure 10.** Three phase current waveform for LG fault.

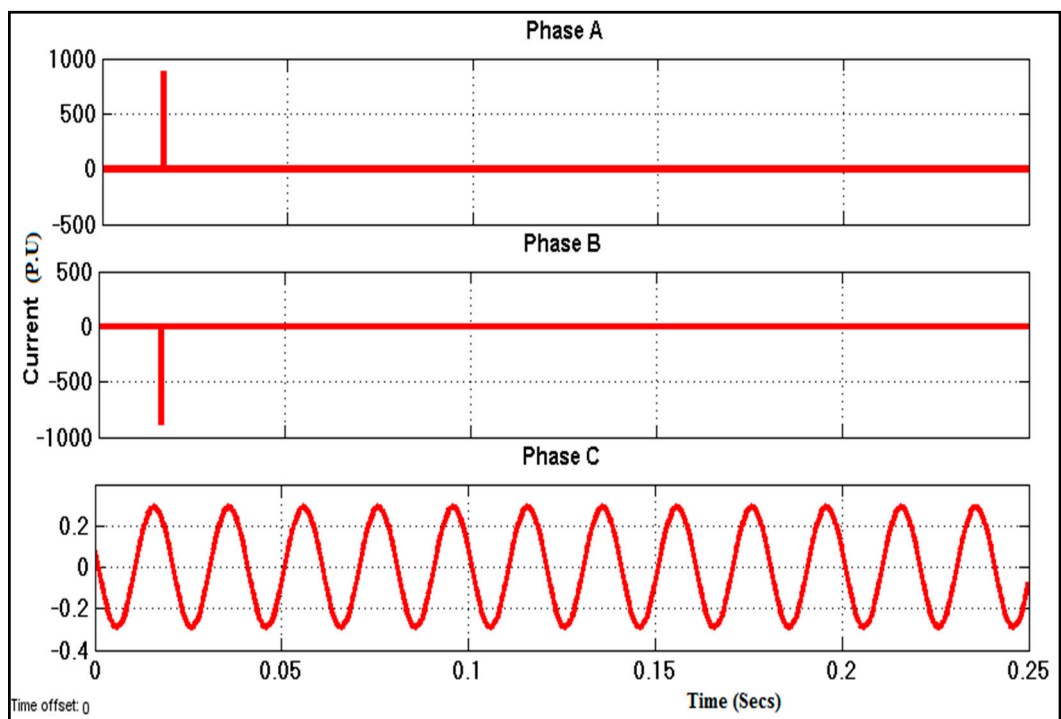

**Figure 11.** Three phase current waveform for LL fault.

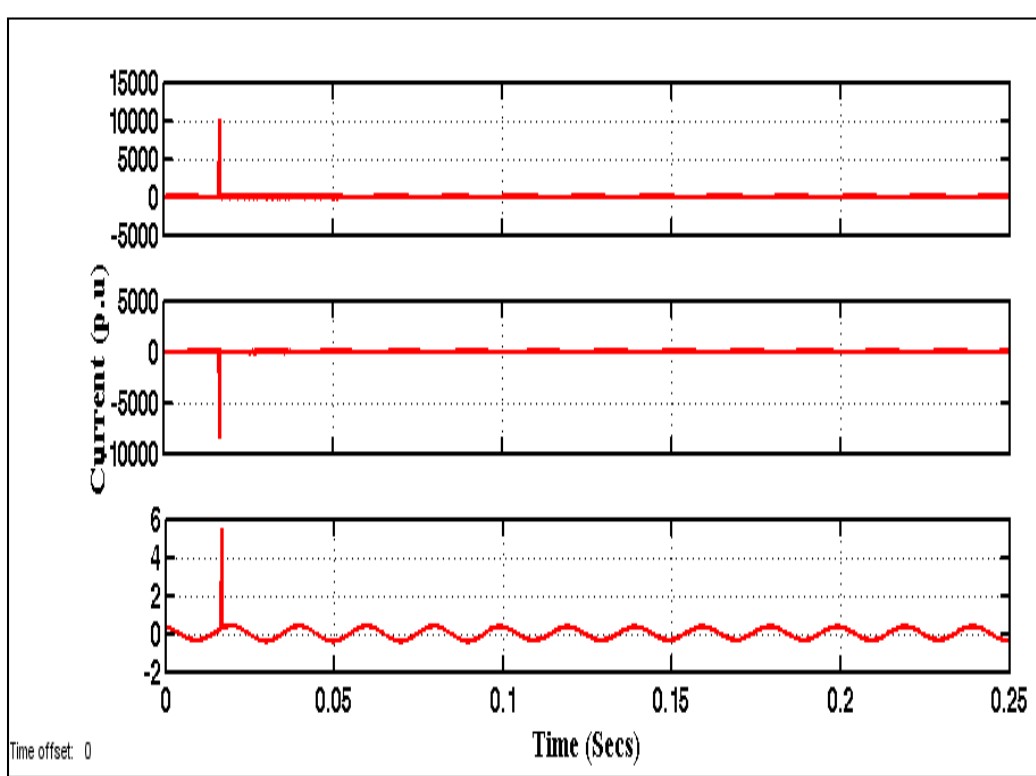

**Figure 12.** Three phase voltage and current waveform for LLG fault.

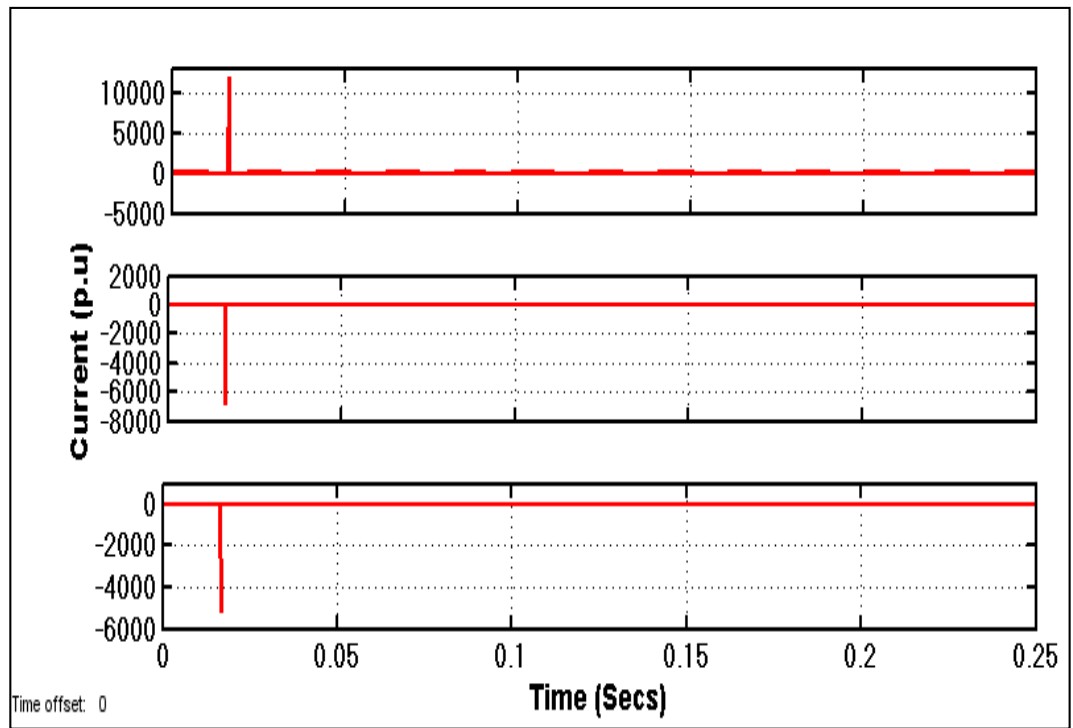

**Figure 13.** Three Phase current waveform for LLLG fault.

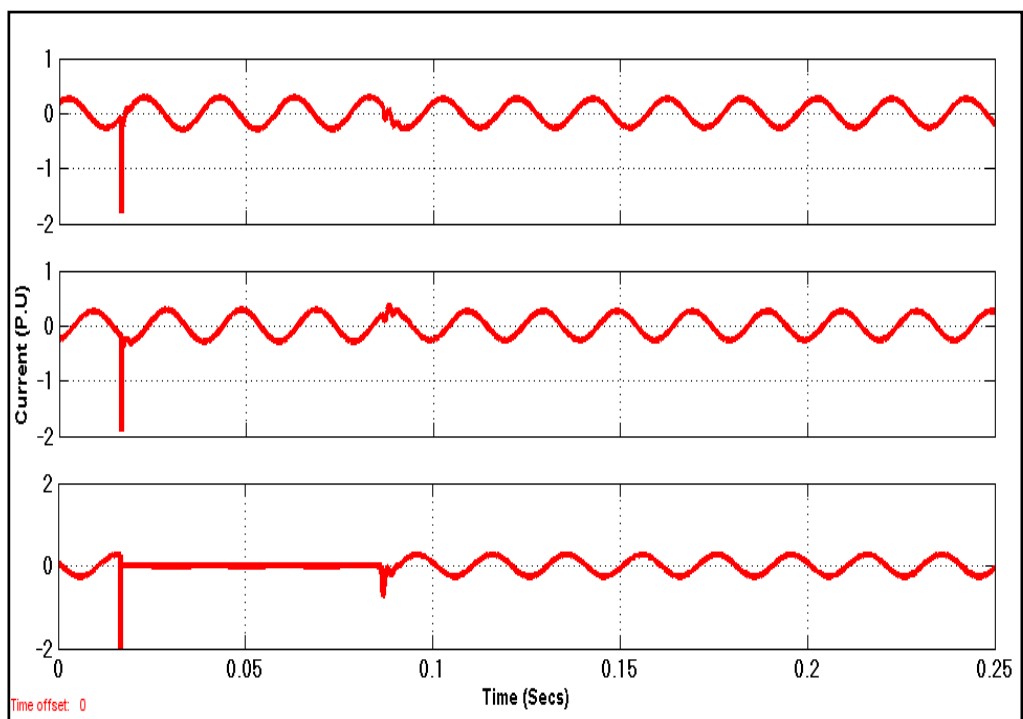

**Figure 14.** Three Phase Current waveform for High Impedance fault at phase C.

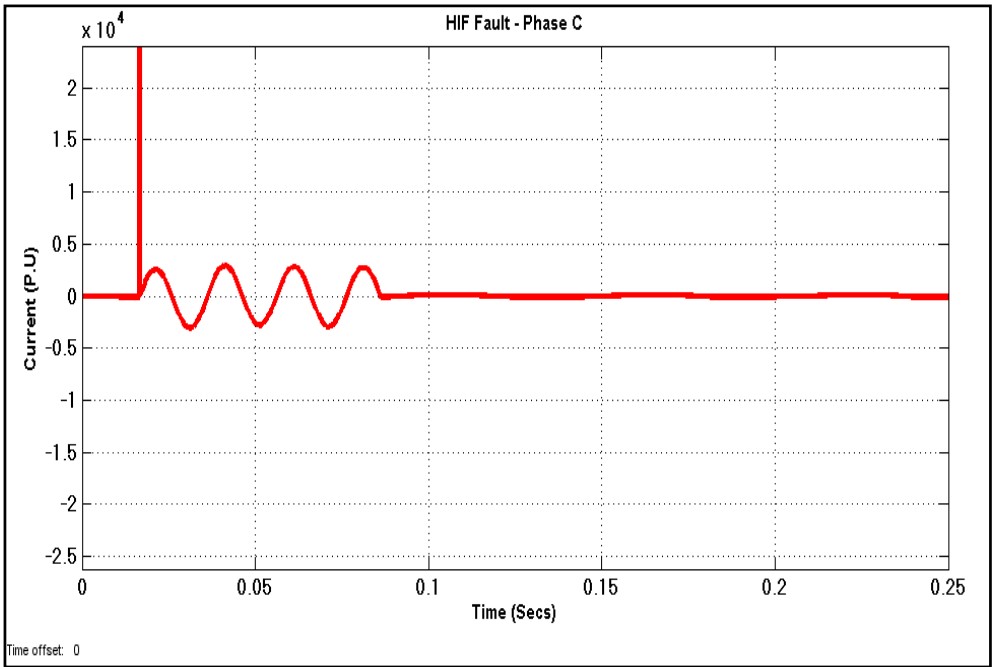

**Figure 15.** High Impedance fault current.

## 5.2. DWT Analysis

The DWT analysis of current signal under different states of distribution system such as Normal, LG, LL, LLG, LLLG and HIF is carried out with the fault resistance of $R_f$ = 0.01 ohm in this section. The  current signal of all phases under Normal operation of the system and also the current signal of faulty phases of different faults is shown from Figures 16–27 respectively for better understanding. It is seen the magnitude of noise presents in the level d1 to d3 is high for all cases of fault and the transients are completely eradicated in the level d4 and d5. A5 is the approximation signal of level d5 and the feature extraction (SD values) obtained using Equation (7) presented in Section 3.1.2 is given in Table 3.

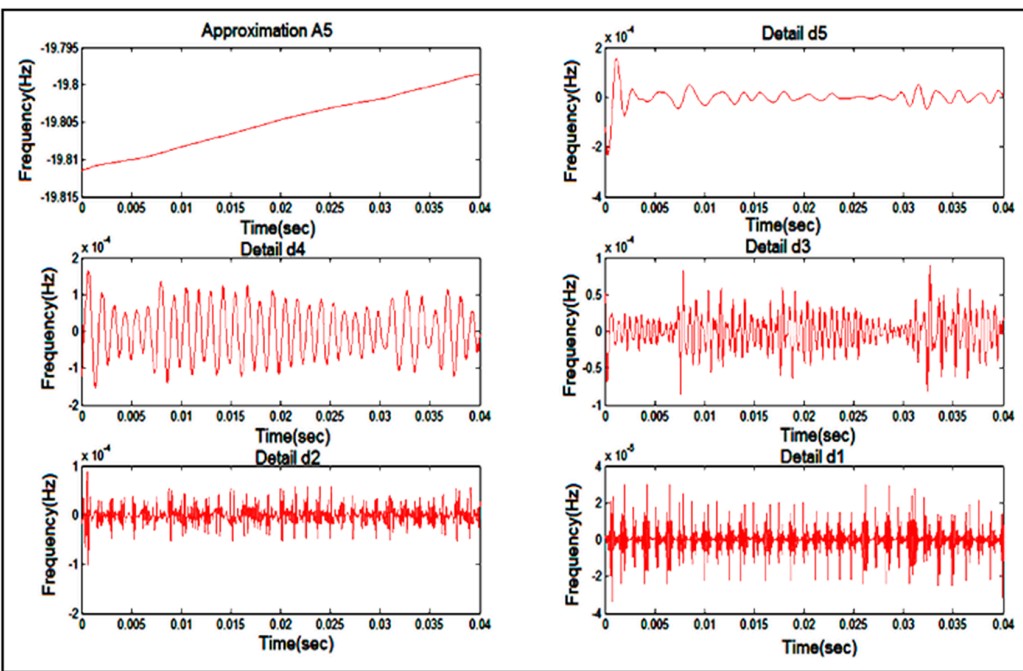

**Figure 16.** DWT analysis of Phase A under Normal Case.

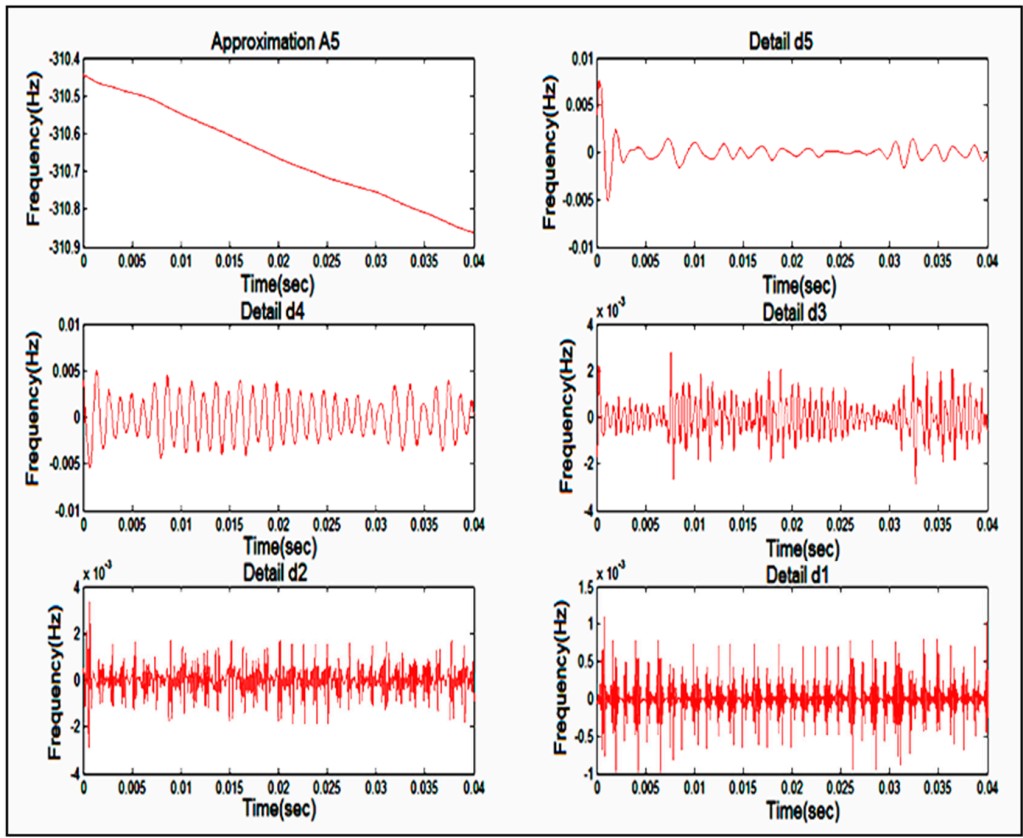

**Figure 17.** DWT analysis of Phase B under Normal Case.

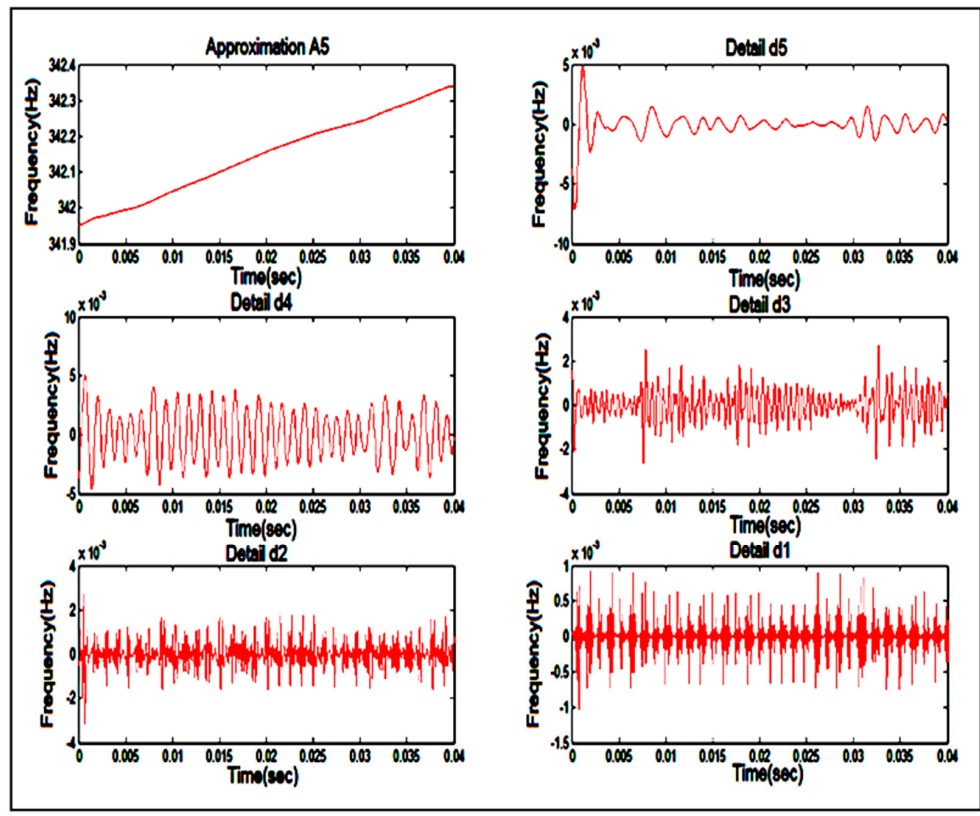

**Figure 18.** DWT analysis of Phase C under Normal Case.

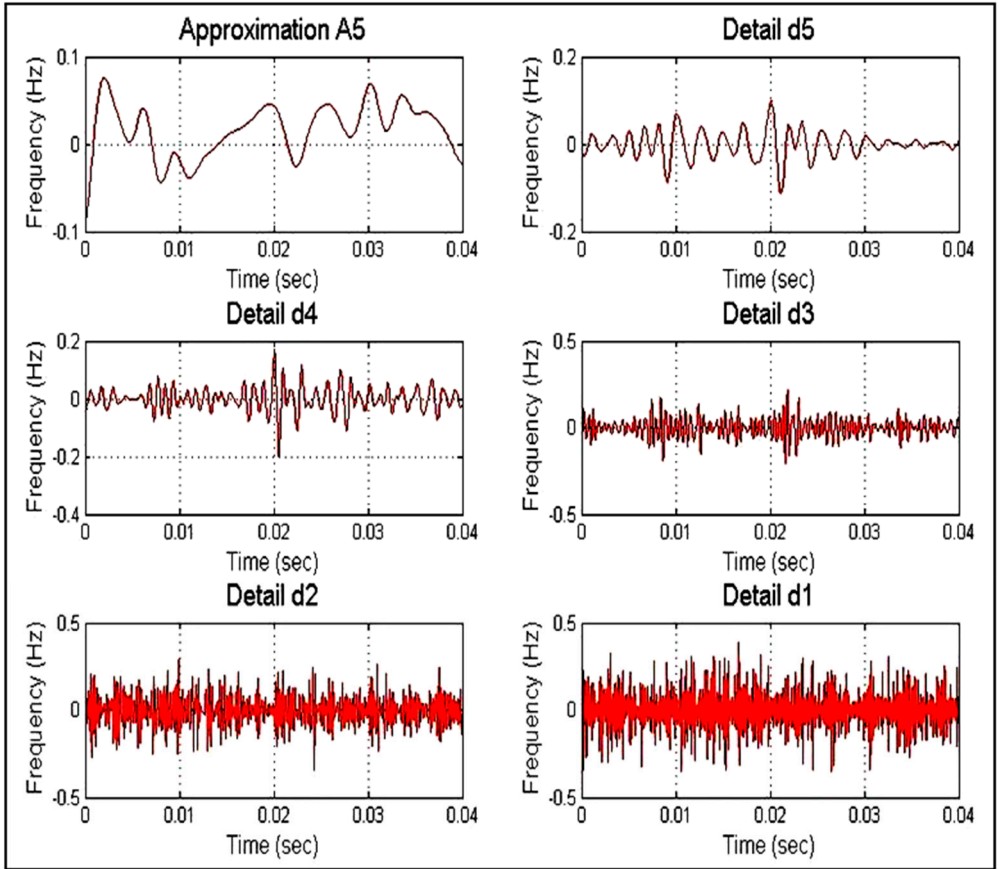

**Figure 19.** DWT waveform of Phase A (LG fault).

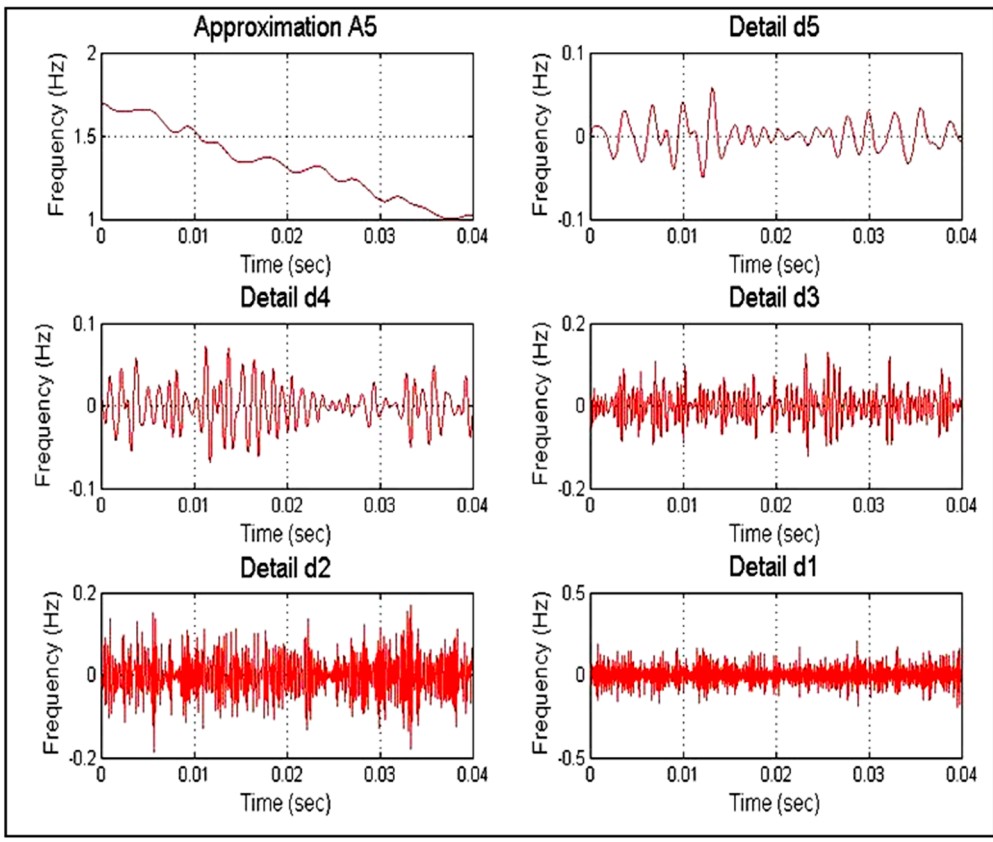

**Figure 20.** DWT waveform of Phase A (LL fault).

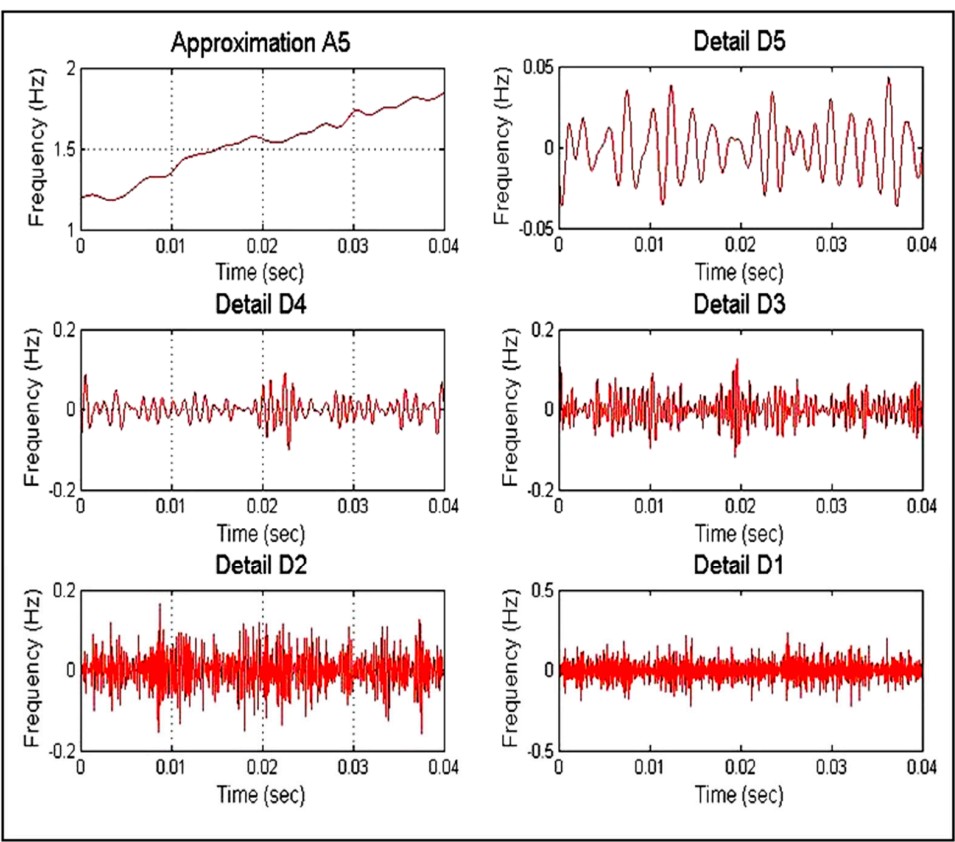

**Figure 21.** DWT waveform of Phase B (LL fault).

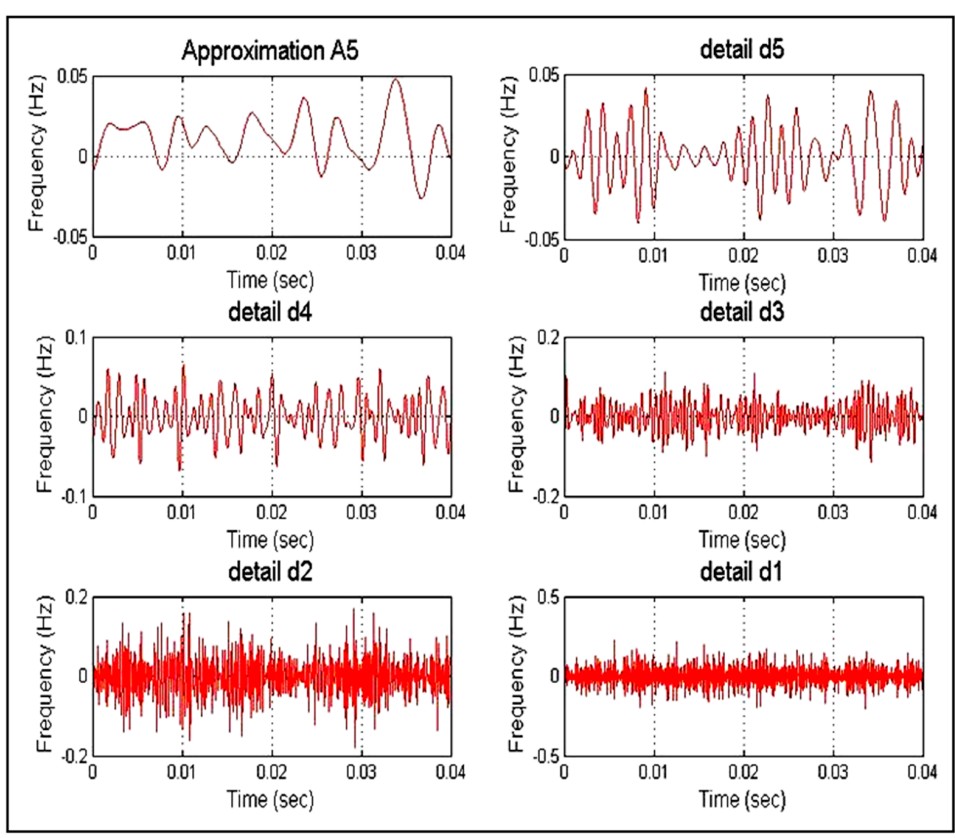

**Figure 22.** DWT waveform of Phase A (LLG fault).

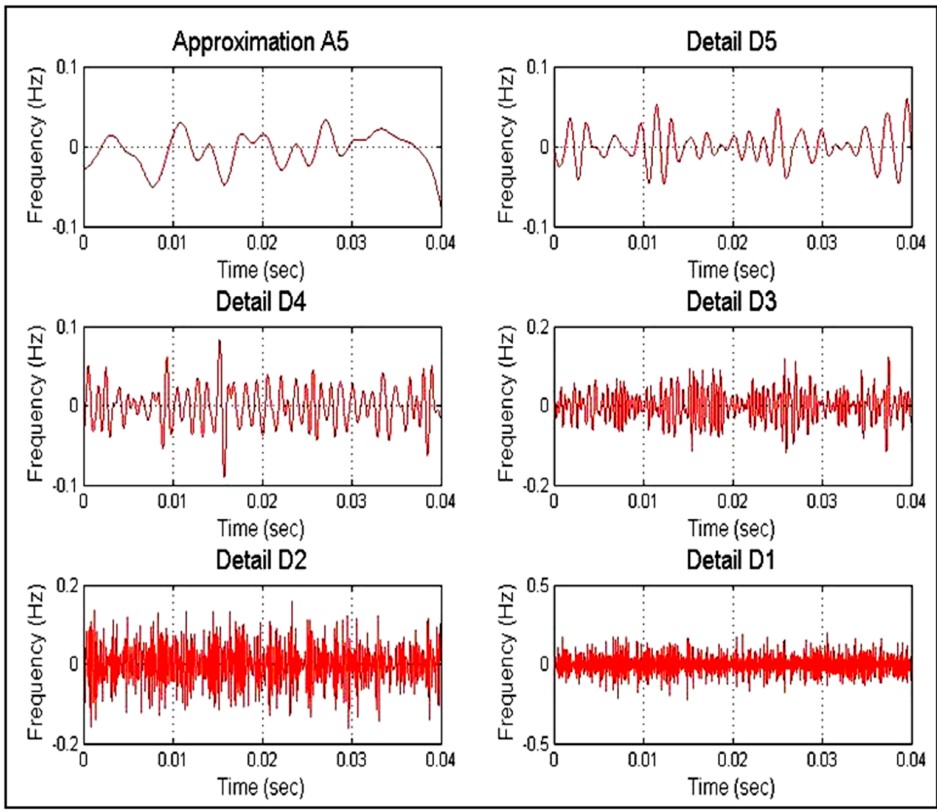

**Figure 23.** DWT waveform of Phase B (LLG fault).

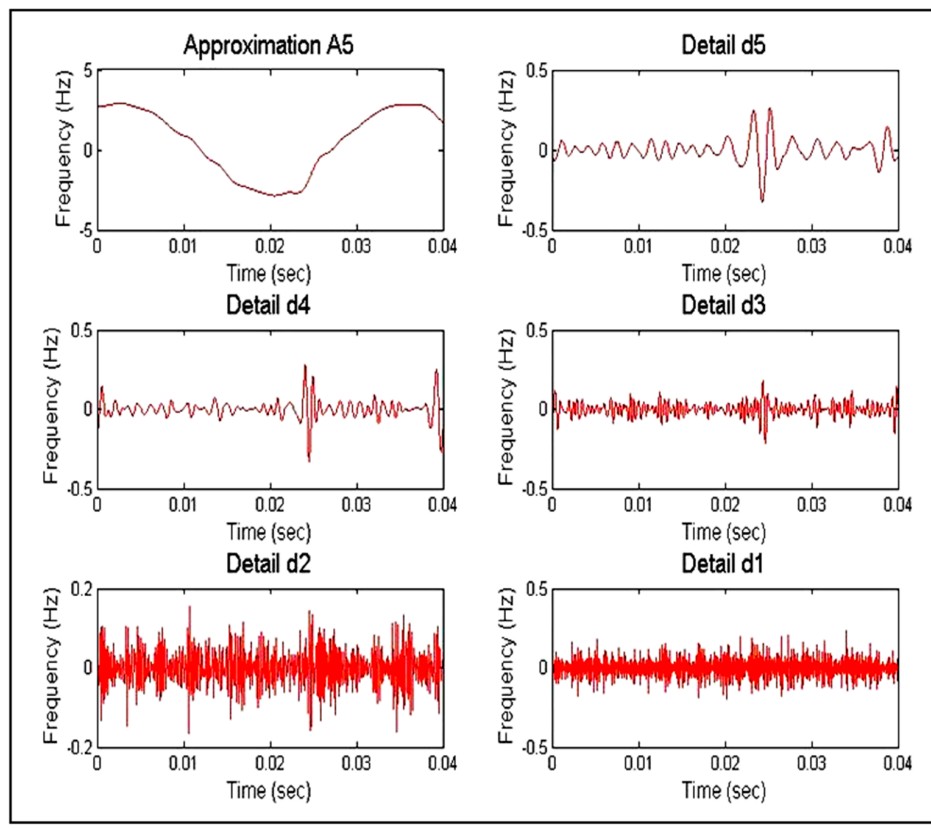

**Figure 24.** DWT waveform of Phase A (LLLG fault).

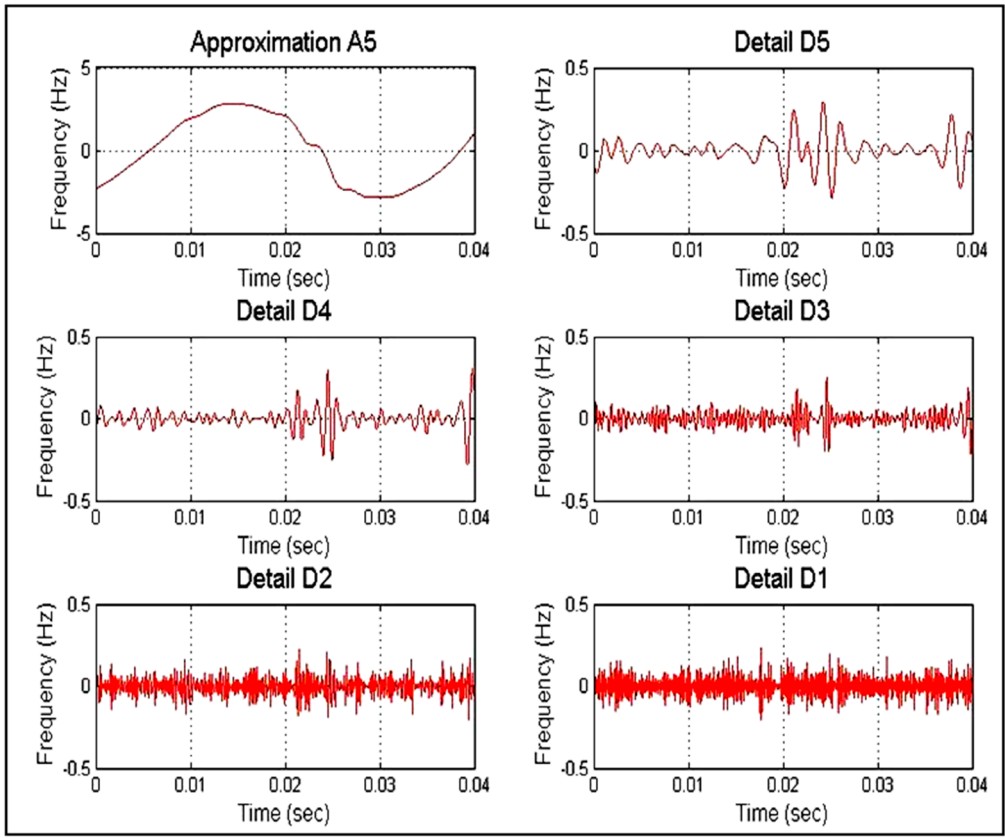

**Figure 25.** DWT waveform of Phase B (LLLG fault).

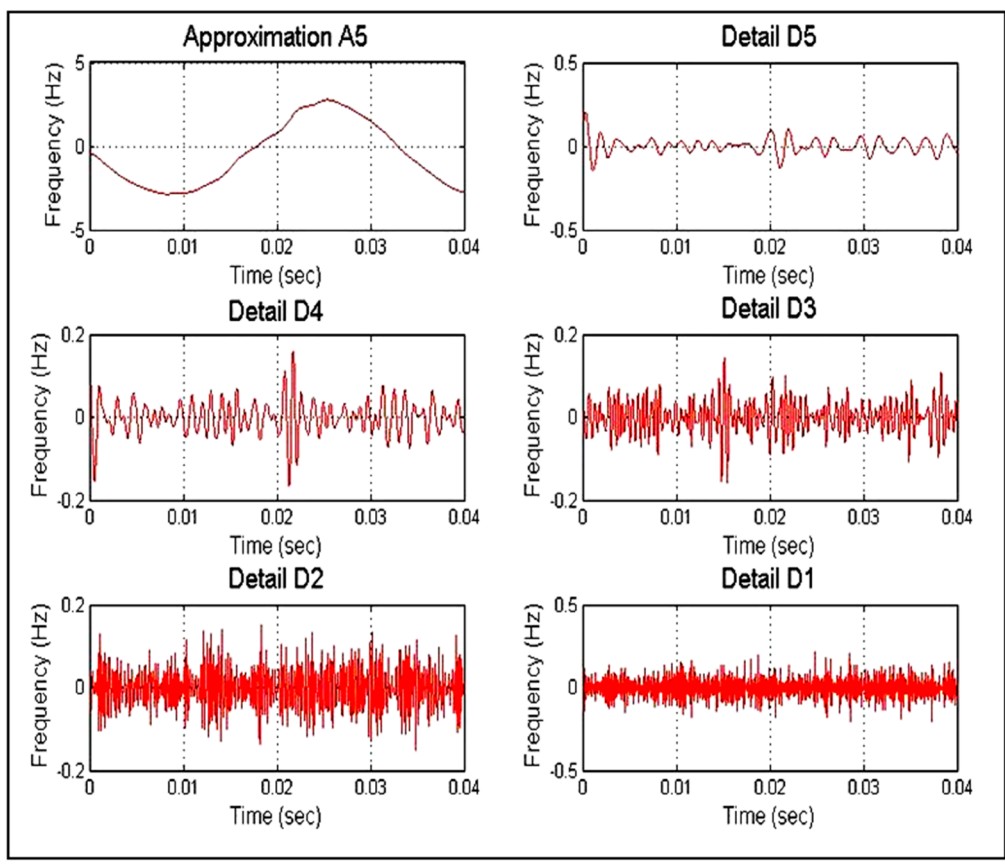

**Figure 26.** DWT waveform of Phase C (LLLG fault).

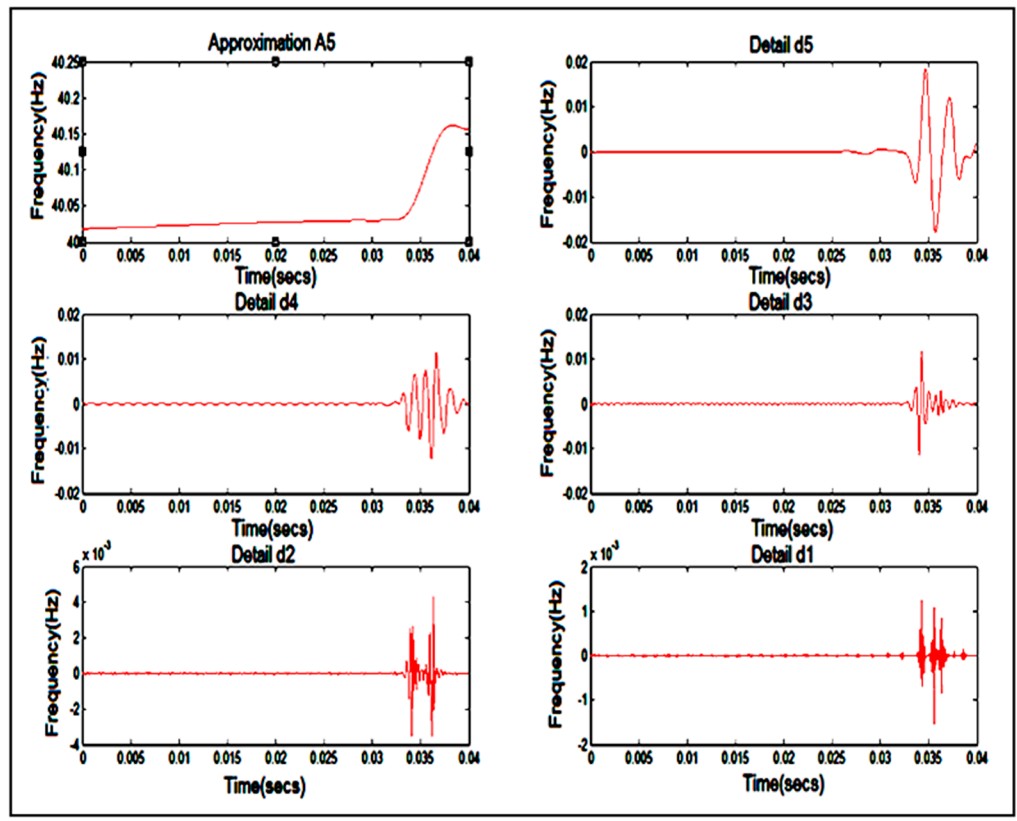

**Figure 27.** DWT waveform of HIF fault at Phase C.

**Table 3.** SD values for current signal with fault resistance ($R_f$ = 0.01ohm).

| Cases | Power System Distribution | SD Values of $I_f$ |
|:---:|:---:|:---:|
| 1 | **Normal Case** | |
| | Phase A | 21 |
| | Phase B | 20.3 |
| | Phase C | 19.0 |
| 2 | **HIF** | |
| | Phase A | 20.33 |
| | Phase B | 19.0 |
| | Phase C | 0.264 |
| 3 | **Three Phase Fault** | |
| | Phase A | 0.3467 |
| | Phase B | 0.3477 |
| | Phase C | 0.341 |
| 4 | **LL Fault** | |
| | Phase A | 0.0113 |
| | Phase B | 0.0132 |
| | Phase C | 0.0105 |
| 5 | **LG Fault** | |
| | Phase A | 0.0127 |
| | Phase B | 0.02 |
| | Phase C | 0.0227 |
| 6 | **LLG Fault** | |
| | Phase A | 0.0115 |
| | Phase B | 0.0143 |
| | Phase C | 0.0255 |

The samples for training the classifier were obtained by varying the fault resistance and the SD values for this case can be obtained similar to SD values with $R_f$ = 0.01 ohm, presented in Table 4.

The rules are framed using the SD values of current signal for different value of fault resistance to train FLS and ANFIS system.

**Table 4.** SD values of current signal for various cases of fault with fault resistance ($R_f$).

| State | Fault with Various $R_f$ | S1 | S2 | S3 | FUZZY Output | Remarks | ANFIS Output | Remarks |
|---|---|---|---|---|---|---|---|---|
| Normal | Normal | 20.33 | 21.22 | 23 | Normal | ✓ | Normal | ✓ |
| 3 Phase Fault | ABC/20 Ohm | 40.33 | 41.54 | 46 | ABC | ✓ | ABC | ✓ |
| | ABC/40 Ohm | 31.38 | 33 | 35.98 | ABC | ✓ | ABC | ✓ |
| | ABC/60 Ohm | 28 | 27.74 | 27 | ABC | ✓ | ABC | ✓ |
| LLG Fault | ABG/20 Ohm | 30 | 34.5 | 23 | ABG | ✓ | ABG | ✓ |
| | ABG/40 Ohm | 29 | 30 | 22.45 | ABG | ✓ | ABG | ✓ |
| | ABG/60 Ohm | 28.42 | 28.88 | 21 | ABG | ✓ | ABG | ✓ |
| | BCG/20 Ohm | 20.03 | 34.76 | 34 | BCG | ✓ | BCG | ✓ |
| | BCG/40 Ohm | 20 | 32 | 31 | BCG | ✓ | BCG | ✓ |
| | BCG/60 Ohm | 19.55 | 27 | 29 | BCG | ✓ | BCG | ✓ |
| | ACG/20 Ohm | 34.45 | 23.33 | 35.1 | ACG | ✓ | ACG | ✓ |
| | ACG/40 Ohm | 32 | 22.3 | 31 | ACG | ✓ | ACG | ✓ |
| | ACG/60 Ohm | 29 | 20 | 28 | ACG | ✓ | ACG | ✓ |
| LG fault | AG/20 Ohm | 40.33 | 23 | 22.64 | AG | ✓ | AG | ✓ |
| | AG/40 Ohm | 35 | 21 | 20.06 | AG | ✓ | AG | ✓ |
| | AG/60 Ohm | 29.98 | 19 | 20 | AG | ✓ | AG | ✓ |
| | BG/20 Ohm | 21 | 47 | 20.06 | BG | ✓ | BG | ✓ |
| | BG/40 OHMS | 18 | 37 | 18.63 | BG | ✓ | BG | ✓ |
| | BG/60 Ohm | 19.73 | 30 | 22 | BG | ✓ | BG | ✓ |
| | CG/20 Ohm | 18.6 | 23 | 47 | CG | ✓ | CG | ✓ |
| | CG/40 Ohm | 19.18 | 22 | 34.98 | CG | ✓ | CG | ✓ |
| | CG/60 Ohm | 21 | 20.87 | 29.61 | CG | ✓ | CG | ✓ |
| LL Fault | AB/20 Ohm | 45.55 | 46.7 | 21 | AB | ✓ | AB | ✓ |
| | AB/40 Ohm | 40 | 37 | 20.1 | AB | ✓ | AB | ✓ |
| | AB/60 Ohm | 34 | 32 | 23 | ABG | ✗ | AB | ✓ |
| | BC/20 Ohm | 21 | 45 | 44 | BC | ✓ | BC | ✓ |
| | BC/40 Ohm | 20.45 | 36 | 37 | BC | ✓ | BC | ✓ |
| | BC/60 Ohm | 24 | 32 | 29.24 | BCG | ✗ | BC | ✓ |
| | AC/20 Ohm | 45 | 23 | 46.9 | AC | ✓ | AC | ✓ |
| | AC/40 Ohm | 35.55 | 22.1 | 36 | AC | ✓ | AC | ✓ |
| | AC/60 Ohm | 32 | 21 | 29 | ACG | ✗ | AC | ✓ |
| HIF Fault | HIF A/75 Ohm | 8 | 21 | 22.2 | HIF A | ✓ | HIF A | ✓ |
| | HIF A/50 Ohm | 11 | 20.09 | 23.4 | HIF A | ✓ | HIF A | ✓ |
| | HIF A/40 ohm | 14.5 | 19 | 24 | NORMAL | ✗ | HIF A | ✓ |
| | HIF B/75 Ohm | 21 | 9 | 20.01 | HIF B | ✓ | HIF B | ✓ |
| | HIF B/50 Ohm | 20.09 | 12.4 | 23.05 | HIF B | ✓ | HIF B | ✓ |
| | HIF B/ 40 Ohm | 19 | 14 | 22 | Normal | ✗ | HIF B | ✓ |
| | HIF C/75 Ohm | 18.76 | 21 | 8.13 | HIF C | ✓ | HIF C | ✓ |
| | HIF C/50 Ohm | 19.61 | 20.19 | 12.09 | HIF C | ✓ | HIF C | ✓ |
| | HIF C/40 Ohm | 20.08 | 19.89 | 15.5 | Normal | ✗ | HIF C | ✓ |

*5.3. Comparative Analysis*

This section describes the comparative analysis of proposed ANFIS method with the FLS approach, the effectiveness of these trained intelligence methods were measured by the success and discriminate rate of each method to identify and distinguish high impedance fault from other power system disturbances. The success and discrimination rate is defined using Equations (8) and (9) as below,

$$Success\,rate = \frac{Number\ of\ HIF\ detected}{Total\ number\ of\ HIF\ events} \cdot 100\% \qquad (8)$$

$$\text{Discrimination rate} = \left[1 - \frac{\text{Number of Events incorrectly diagnosed}}{\text{Total number of events}}\right] \cdot 100\% \tag{9}$$

The results obtained from FIS and ANFIS are shown in Table 4, which shows that the success rate and discrimination rate of FIS is 66.67% and 85% respectively. On the other hand, the ANFIS method of classification has the success and discrimination rate of 100% which proves to be a more effective method of classifying the HIF fault than the FIS system.

## 6. Conclusions

In this research, a medium voltage distribution network of 13.8 kV has been simulated using MATLAB (R14)/Simulink (5.0) by applying various types of faults in the feeder of distribution network. The current waveform obtained in each case of normal, symmetrical, asymmetrical and high impedance fault (HIF) has been analyzed using Discrete Wavelet Transform (DWT) of Db9 mother wavelet in order to locate the type of fault in the distribution system. The signal has been sampled using DWT with different band of frequencies which is represented as 1st, 2nd, 3rd, 4th, and 5th level of detailed coefficient and approximation level of 5 of current signal. The SD values obtained from DWT analysis of each case has been used to classify the type of fault occurred in the feeder network. To classify the type of fault, wide amount of data can be obtained by simulating the network with different values of fault resistance such as 0.01, 40, 50, and 70 ohm which is used to train the FLS and ANFIS classifier. It is observed from the results that the FLS method of classification has led to misclassification in the case of HIF and LL fault and even some times has shown the HIF condition to be normal condition. But, ANFIS method of classification has given an appropriate solution in case of misclassification. The success and discrimination rate of FLS method have been 66.67% and 85% where as ANFIS method of classification can provide 100% of success and discrimination rate. Hence, it is concluded that the ANFIS method of fault classification is 33.37% more effective to identify the high impedance fault and 15% effective in overall identifying the type of disturbances that occur in radial distribution system than FLS.

**Author Contributions:** V.V. proposed the main idea and performs the simulation of the work; V.V. and N.I.A.W. wrote this paper; R.R. provided sources and assisted to write paper; M.T. proposed Fuzzy approach; proof reading and final drafting was done by M.M. and M.L.O.

**Funding:** This research work was supported in part by the GPB-UPM under Grant No. 9630000.

**Acknowledgments:** The authors are thankful to center for Advanced Lightning and Power Energy System (ALPER) and Universiti Putra Malaysia (UPM) for carrying out the research.

**Conflicts of Interest:** The authors declare no conflict of interest.

## Abbreviation

| Variables | Explanation |
|---|---|
| D1 to D5 | Detailed coefficients of level 1 to 5 |
| A5 | Approximate coefficients of level 5 |
| LG | Line to ground fault |
| LL | Line to Line fault |
| LLG | Double line to ground fault |
| LLLG | Three phase fault |
| HIF | High impedance fault |
| SD | Standard Deviation |
| Db9 | Daubichies's mother wavelet |

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
