# Peer review of "High Impedance Fault Detection in Medium Voltage Distribution Network Using Discrete Wavelet Transform and Adaptive Neuro-Fuzzy Inference System"

_energies, doi:10.3390/en11123330_

Round 1
Reviewer 1 Report
This paper presents a method to detect and classify the high impedance fault that occur in the medium voltage distribution network using discrete wavelet transform (DWT) and adaptive neuro-fuzzy inference system (ANFIS). The results indicate that the proposed method is more efficient to identify and discriminate the high impedance fault from other power system faults in the system.
As a general comment, I think the paper makes an interesting contribution to the literature by providing this new method. At the same time, I think the paper needs very minor improvements before to be published in this journal.
Broad comments
1) In the introduction the authors make a good description of different methods used to identify high impedance fault. I would suggest the author to mention also the following articles where in particular in the second a comparison about different methods is presented:
a. Rafał Weron,“Electricity price forecasting: A review of the state-of-the-art with a look into the future”, International Journal of Forecasting, Volume 30, Issue 4, Pages 1030–1081, 2014
b. Silvano Cincotti, Giulia Gallo, Linda Ponta, Marco Raberto, “Modelling and forecasting of electricity spot-prices: Computational intelligence vs classical econometrics”, AI Communications, Volume 27, Issue 3, Pages 301-314, 2014
2) In subsection 3.1.1, the choice of 5000 points is not clear. Please justify.
3) In order to let the paper more readable, I would suggest the authors to add a table with the main variables used.
4) Please make all the figures with the same style.
Specific comments
Line 26, please remove “accurately”, or modify the sentence
Line 200-201: the sentence is not clear. Please modify.
Author Response
Reviewer# :1
This paper presents a method to detect and classify the high impedance fault that occur in the medium voltage distribution network using discrete wavelet transform (DWT) and adaptive neuro-fuzzy inference system (ANFIS). The results indicate that the proposed method is more efficient to identify and discriminate the high impedance fault from other power system faults in the system.
As a general comment, I think the paper makes an interesting contribution to the literature by providing this new method. At the same time, I think the paper needs very minor improvements before to be published in this journal.
Broad comments:#
1) In the introduction the authors make a good description of different methods used to identify high impedance fault. I would suggest the author to mention also the following articles where in particular in the second a comparison about different methods is presented:
a. Rafał Weron,“Electricity price forecasting: A review of the state-of-the-art with a look into the future”, International Journal of Forecasting, Volume 30, Issue 4, Pages 1030–1081, 2014
b. Silvano Cincotti, Giulia Gallo, Linda Ponta, Marco Raberto, “Modelling and forecasting of electricity spot-prices: Computational intelligence vs classical econometrics”, AI Communications, Volume 27, Issue 3, Pages 301-314, 2014.
As recommended, the aforementioned manuscript was added in the reference list and the work is cited in the manuscript.
2) In subsection 3.1.1, the choice of 5000 points is not clear. Please justify.
The value 5000 points is the number of samples or length of fault current signal analyzed. In this work, the number of samples is chosen random for analysis.
3) In order to let the paper more readable, I would suggest the authors to add a table with the main variables used.
As recommended, the main variables used in the manuscript were presented in Table 1.
4) Please make all the figures with the same style.
As recommended, all the figures in the manuscript are changed to same style.
Specific comments:#
Line 26, please remove “accurately”, or modify the sentence
As recommended, the word “accurately” is removed.
Line 200-201: the sentence is not clear. Please modify.
As recommended, the section 3.1.2 is completely revised for better understanding.

Reviewer 2 Report
The authos should add into more citations about discrete wavelet transform (DWT) and adaptive neuro-fuzzy inference system in Energies Journal.
Equation (4) is lost.
The block diagram of Fig. 8 should be explained.
The English should be improved.
The Authors should revise the contents according to the following comments. 1. The Authors should explain and list the different between the discrete wavelet transform (DWT) and the other methods. 2. The system equations should be listed in detail. 3. English syntax should be improved. 4. The Authors should list the different between the other neural networks and neuro-fuzzy inference system (ANFIS).Author Response
Reviewer#:2
The authors should add into more citations about discrete wavelet transform (DWT) and adaptive neuro-fuzzy inference system in Energies Journal.
As suggested, the references based on discrete wavelet transform (DWT) and adaptive neuro-fuzzy inference system (ANFIS) was cited from Energies Journal.
Equation (4) is lost.
The general form of decomposition of signal for N-level using DWT is represented as equation (4) and it is given below,
The block diagram of Fig. 8 should be explained.
The HIF fault model designed using Matlab as depicted in Figure 8 consists of saw tooth current controller, constant resistor, variable resistor of non-linear fashion and diodes. The developed model has better dynamic arc current characteristics which depicts the non-linearity of ground resistance.
The Authors should revise the contents according to the following comments.
1. The Authors should explain and list the different between the discrete wavelet transform (DWT) and the other methods.
The discrete wavelet transform (DWT) is a powerful tool which allows the analysis of fault current signal localized both in time and frequency. On the other hand, the transform theory such as Fast Fourier Transform (FFT) and Short Time Fourier Transform (STFT) technique gives the information about the signal localized only in frequency, the time at which particular disturbance or fault occurred in the signal is lost. Therefore, to overcome such drawbacks wavelet analysis were used in the proposed work to localize the signal both in time and frequency simultaneously which helps to detect the time of occurrence of fault or disturbances effectively.Thus the wavelet transform can be analyzed in two ways: Continuous Wavelet Transform (CWT) and Discrete Wavelet Transform (DWT). Among these two methods, the latter is extensively used because of the following reasons,
CWT requires a large number of scales to show the signal components, which makes it useless for online application.
CWT is highly redundant transform as its wavelet coefficients contain more information than necessary.
CWT provides the region where the fault occurs, but DWT localize the fault more efficient.
DWT preserve all the information of the function with minimum number of wavelet coefficients.
Computational time is faster for DWT analysis.
Construction of CWT inverse is more difficult.
Therefore in this proposed work the DWT method of signal processing technique was used for feature extraction to locate the high impedance fault in the power system.
2. The system equations should be listed in detail.
In the proposed work, the feeder network is modelled using Matlab/Simulink and the fault current signal is obtained from simulation for analysis using DWT technique. The concept of DWT technique were detailed in the manuscript with the aid of fault current signal as x(t) and it is described in the form of equations as in section 3.1.
Since the system is modelled and simulated, the fault current is not obtained from the mathematical equation, therefore the system equation were not presented in the manuscript.
3. English syntax should be improved.
As recommended, English syntax was verified.
4. The Authors should list the different between the other neural networks and neuro-fuzzy inference system (ANFIS).
Adaptive Neuro Fuzzy Interface System (ANFIS) is one of the greatest tradeoff among ANNs and fuzzy logic systems, offering smoothness due to the fuzzy control interpolation and adaptability due to the ANN back propagation. ANFIS provide a technique for the implementation of fuzzy inference system to adaptive networks for developing fuzzy rules with proper membership functions to have required inputs and outputs. An adaptive network is a feed-forward multi-layer neural network with adaptive nodes in which the outputs are predicted on the parameters of the adaptive nodes and the adjustment of parameters due to error term is specified by the learning rules. ANFIS is a class of ANN, which incorporates both ANN and fuzzy logic principles and has benefits of both techniques in a single framework as follows:
·It is capable of handling complex and nonlinear problems even if the targets are not given.
· The learning duration of ANFIS is very short than NN which implies that ANFIS reaches the target faster than neural network.
· Reduces the complexity of the problem, in case of system with huge amount of data.
·In training of the data, ANFIS gives result with minimum total error compared to other type of NN.
